# EKKA: Automated Diagnosis of Silent Errors in LLM Inference

Yile Gu [1]  Zhen Zhang [2]  Shaowei Zhu [2]  Xinwei Fu [2]  Jun Wu [2]  Yida Wang [2]  Baris Kasikci [1]

## Abstract

LLM serving frameworks are quickly evolving with a complex software stack and a vast number of optimizations. The rapid development process can introduce silent errors where output quality silently degrades without any explicit error signals. Diagnosing silent errors is notoriously difficult due to the substantial semantic gap between the high-level symptoms and the low-level root causes. We observe that diagnosis of silent errors can be effectively framed as a differential debugging problem by leveraging the existence of semantically correct reference implementations. We propose EKKA, an automated diagnosis system that identifies root causes by systematically aligning and comparing intermediate execution states between a target and a reference framework. We constructed a benchmark of real-world silent errors from popular serving frameworks, where EKKA shows 80% pass@1 diagnosis accuracy and 88% pass@5 diagnosis accuracy, outperforming state-of-the-art systems. EKKA also diagnoses 4 new silent errors from serving frameworks, all of which have been confirmed by the developers.

## 1. Introduction

Large Language Model (LLM) inference has emerged as a critical workload powering a vast number of downstream applications, from interactive chatbots to complex reasoning agents (Ouyang et al., 2022; Park et al., 2023; Wang et al., 2023). To meet the stringent latency and throughput demands of these applications deployed in production, LLM inference increasingly relies on dedicated serving frameworks that are efficient and performant.

LLM serving frameworks have evolved into complex, highly optimized systems. To maximize efficiency, these frameworks typically incorporate sophisticated optimizations like

| Input | Question: Rory orders 2 subs for \$7.50 each, 2 bags of chips for \$1.50 each and 2 cookies for \$1.00 each for delivery. There's a 20% delivery fee and a \$5.00 tip. What will her order cost? |
|---|---|
| **HuggingFace** *(Reference)* | The cost of the subs is $2 \times \$7.50 = \$15$. The cost of the chips is $2 \times \$1.50 = \$3$. The cost of the cookies is $2 \times \$1.00 = \$2$. The subtotal is $\mathbf{15 + 3 + 2 = 20}$. The delivery fee is $20\%$ of $20 = 4$. **Total: \$29** |
| **vLLM** *(Buggy)* | The cost of the subs is $2 \times \$7.50 = \$15$. The cost of the chips is $2 \times \$1.50 = \$3$. The cost of the cookies is $2 \times \$1.00 = \$2$. The total cost of the food is $15 + 15 + 15 = 45$. The delivery fee is $45 \times 0.20 = 9$. **Total: \$100** |

*Figure 1.* The output comparison from HuggingFace and vLLM for the same prompt for the silent error vLLM-17689.

paged attention, radix attention, and custom CUDA kernels (Kwon et al., 2023; Zheng et al., 2024; Zhu et al., 2025; Zhong et al., 2024; Chen et al., 2025). While these optimizations deliver performance gains, the increasing complexity of the serving stack makes these frameworks highly susceptible to software defects (Yu et al., 2025a; Chen et al., 2023; Liu et al., 2026).

This inherent complexity in the frameworks frequently leads to silent errors that are distinct from traditional crash-inducing failures. Unlike failures that produce explicit error signals such as runtime errors or assertion failures, silent errors allow the serving framework to process requests and return responses without error, while the output quality silently degrades. These symptoms range from nonsensical outputs, malformed structures to subtle benchmark regressions.

Figure 1 shows the symptoms of a recent silent error in vLLM (Kwon et al., 2023) that caused the Gemma 3 model's (Team et al., 2025) accuracy on the Hellaswag (Zellers et al., 2019) benchmark to drop nearly 30% without triggering any runtime errors or warnings. While the inference engine remained operational, it produced plausible yet incorrect outputs, leading developers to spend months misdiagnosing the issue before identifying a subtle sliding window attention misuse deep in the model stack.

Diagnosing such silent errors is notoriously difficult due to

---

[1]University of Washington [2]Amazon Web Services. Correspondence to: Baris Kasikci <baris@cs.washington.edu>.

*Proceedings of the 43rd International Conference on Machine Learning*, Seoul, South Korea. PMLR 306, 2026. Copyright 2026 by the author(s).

the substantial semantic gap between the high-level symptom and the low-level root cause. Our analysis of silent errors in LLM serving frameworks reveals that root causes are diverse across the serving stack, from framework-level implementation to kernel optimizations. Existing fault localization techniques typically depend on explicit pass/fail signals, which are absent for silent errors. Deep learning testing tools either act as black-box detectors or restrict comparisons to APIs, making it difficult to isolate root cause inside optimized serving engines. General-purpose agentic debugging tools lack the domain-specific scaffolding needed to diagnose silent errors, leading to ineffectiveness in diagnosis. Consequently, developers are forced to rely on laborious manual diagnosis workflows to diagnose such silent errors.

We observe that diagnosis of silent errors can be effectively framed as a differential debugging problem by leveraging the existence of semantically correct reference implementations. While manually comparing results against a reference (e.g., HuggingFace Transformers (Wolf et al., 2019)) is a common debugging strategy, automating this process is nontrivial because optimized serving frameworks use vastly different internal component structures and memory layouts than reference models. A direct tensor comparison is often impossible without significant manual effort to align the intermediate states of disparate implementations.

To address these challenges, we propose EKKA, an automated diagnosis system that identifies root causes by systematically aligning and comparing intermediate execution states between a target and a reference framework. Our key insight is that with the right scaffolding, LLM agents can effectively recognize implementation differences and align intermediate outputs, thus automating the otherwise laborious differential diagnosis process. EKKA employs a multi-stage agentic workflow that first analyzes the codebase and model architecture, maps semantically equivalent components despite implementation disparities, and generates executable code to align output activations. Finally, EKKA utilizes change-point analysis on a robust error ratio metric that tolerates minor numerical instability to pinpoint the buggy component responsible for the silent error.

Our evaluation on a benchmark of real-world silent errors demonstrates that EKKA effectively localizes root causes with high accuracy and low cost. We successfully diagnosed 17 issues from vLLM (Kwon et al., 2023) and SGLang (Zheng et al., 2024). EKKA shows 24% to 34% improvement on diagnosis accuracy compared to state-of-the-art systems with an average diagnosis cost of approximately $30 per case. EKKA also diagnosed 4 new silent errors from vLLM and SGLang, all of which are confirmed by the developers.

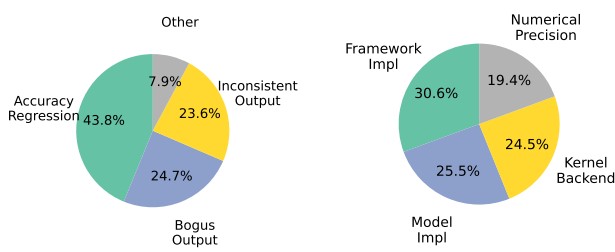

(a) Bug Symptom Distribution    (b) Root Cause Distribution

*Figure 2.* Bug symptoms and root causes of silent errors in vLLM and SGLang.

## 2. Silent Error Study

### 2.1. Bug Collection Methodology

To understand the characteristics of silent errors and to gain insights from how developers diagnose such bugs, we conducted a comprehensive empirical study of real-world issues in LLM serving systems. We selected vLLM (Kwon et al., 2023) and SGLang (Zheng et al., 2024) as our target subjects, as they represent two of the most popular open-source high-performance serving frameworks in use.

Our collection process combined keyword search and manual verification. We retrieved GitHub issues labeled or titled as "bug" and matched quality-regression keywords (e.g., "accuracy", "inconsistent", "garbage"), and further inspected associated PRs for closed issues to understand resolutions. After collecting all candidate issues, we manually inspected them to exclude irrelevant reports. This produced 90 silent errors in total: 48 from vLLM (33 closed, 15 open) and 42 from SGLang (37 closed, 5 open); we use the 70 closed issues for the study and the open ones to evaluate EKKA.

### 2.2. Bug Symptoms

Based on our analysis of 70 collected issues, we classify the bug symptoms into three categories: accuracy regression, inconsistent output, and bogus output. Accuracy regression refers to scenarios where the model generates superficially valid text, but performance on standard benchmarks (e.g., MMLU (Hendrycks et al., 2020), GSM8K (Cobbe et al., 2021)) degrades compared to a baseline. Bogus output involves the generation of nonsense, repetitive loops, or gibberish, while inconsistent output occurs when the framework produces different results for identical inputs across frameworks or configurations. Other symptoms include malformed JSON and broken tool calls that disrupt downstream parsing. Figure 2a shows the distribution of bug symptoms. We observe that accuracy regression is the most prevalent symptom, accounting for 43.8% of all reported issues.

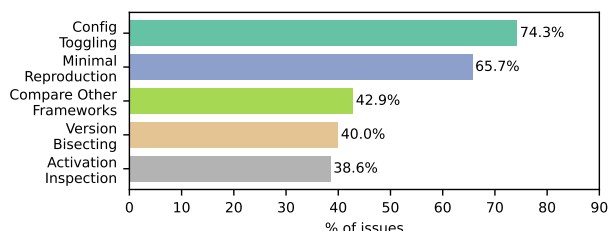

*Figure 3.* Diagnosis actions for silent errors in vLLM and SGLang.

## 2.3. Root Causes

Figure 2b shows the root cause distributions of the silent errors studied, which could be classified into four distinct categories: framework implementation, model implementation, kernel backend, and numerical precision. Framework implementation issues constitute the largest category (30.6%), involving logic errors within the serving engine itself, such as async engine implementation and CUDA graph compilation. The model implementation category (25.5%) are errors in defining model architectures or mis-configuring models with parameters and certain chat templates. Kernel backend bugs (24.5%) arise within specialized compute kernels such as FlashAttention (Dao et al., 2022). Interestingly, around 19.4% of issues are related to numerical precision, which stem purely from floating-point instability (e.g., BF16 accumulation errors) in the absence of logical defects.

## 2.4. Diagnosis Actions

Figure 3 summarizes representative diagnosis actions developers perform during the diagnosis of the 70 resolved accuracy bugs studied. Developers rely on five primary diagnosis actions that progressively isolate the fault from the system's complexity. Configuration toggling (i.e., switching configurations in a framework) and minimal reproduction (i.e., writing unit tests with simplified setup for bug reproduction) are the most ubiquitous strategies, appearing in over 60% of the analyzed issues. Furthermore, comparing with other frameworks is utilized in approximately 50% of cases, particularly within vLLM, which performs differential debugging against a trusted reference framework. These steps are often accompanied by activation inspection to trace tensor values and version bisecting to identify the specific commit that introduced the regression.

## 2.5. Implications

**Substantial semantic gap between symptoms and root causes.** Silent errors often appear as end-to-end accuracy regressions (43.8% of issues), but the root cause may lie anywhere in the framework stack; output-only observation provides little signal without intermediate-state inspection.

**Diverse root causes across different levels in the model stack.** About 50% of silent errors originate in model/kernel implementation rather than high-level orchestration, so diagnosis must be model-aware rather than treating the model as a black box.

**Diagnosis is manual and time-consuming.** Developers compare against a reference framework in about 50% of cases, but aligning intermediate tensors across heterogeneous implementations is labor-intensive, motivating automated mapping and alignment.

## 3. EKKA Design

### 3.1. Workflow Overview

We propose EKKA, an automated system for diagnosing silent errors in LLM serving frameworks via differential debugging. Our key insight is that while optimized serving engines (e.g., vLLM) may contain defects, a semantically correct reference implementation (e.g., HuggingFace Transformers) often exists and can serve as an oracle. Given a buggy target framework and a reference framework, EKKA takes as input the model, benchmark prompts, and configurations on two frameworks, and outputs a ranked report of the suspected component responsible for the divergence.

EKKA runs in two stages. (1) *Diagnosis Information Collection* parses both codebases and model architectures, and collects execution traces (activations and call sequences) while reproducing the bug. (2) *Agent-based Bug Diagnosis* localizes the silent error through three steps: component mapping, activation alignment, and error analysis.

First, in component mapping, EKKA identifies semantically equivalent sub-modules across frameworks despite different module boundaries and naming. Then, in activation alignment, EKKA processes the collected tensors from these mapped component pairs, handling differences in shapes, data types, or memory layouts. Finally, in error analysis, EKKA computes a robust error ratio to separate true defects from numerical noise, and applies change-point analysis on these error ratios to identify the precise moment where divergence spikes, pinpointing the root cause. EKKA targets silent errors rooted in the model stack (model implementations and kernel backends), replacing ad-hoc tensor dumping with a systematic automated pipeline.

### 3.2. Component Mapping

Implementation of the same model could be very different across frameworks, making it hard to automate differential diagnosis. For example, vLLM typically fuses query, key and value projections in the attention module into a unified `QKVProjection` class, while HuggingFace implements them as 3 separate modules. Consequently, naively matching class names of a framework to the other fails easily.

EKKA performs component mapping by combining model-

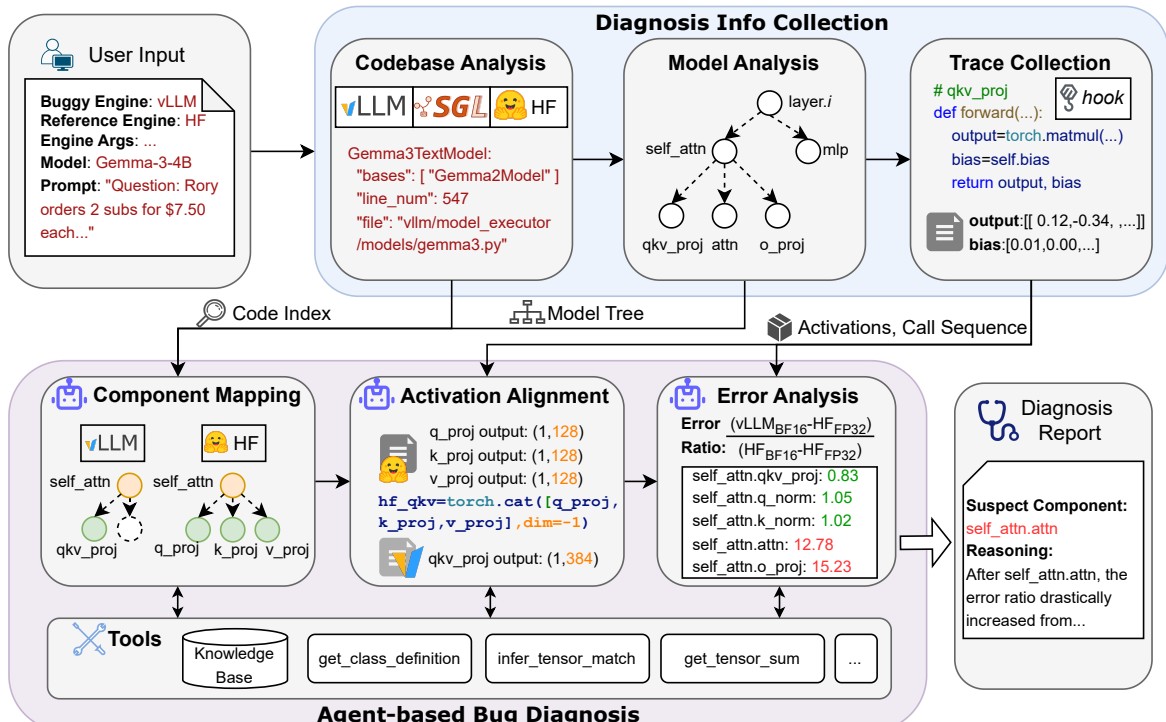

*Figure 4.* The overall architecture of EKKA.

architecture analysis and code inspection. It first builds a *Model Tree* that compresses the module architecture into a concise and hierarchical representation, and then maps semantically equivalent nodes between the target and reference trees. During mapping, EKKA can query `get_class_definition` to inspect the underlying implementation when names are ambiguous, and outputs mapping pairs as one-to-many or many-to-one correspondences. Components that exist in only one framework (e.g., SGLang's logit processor) are left unmapped with an explicit reason, to keep the mapping both correct and complete.

```
root: Gemma3ForConditionalGeneration
|-- embed_tokens: Gemma3WordEmbedding
|-- language_model: Gemma3TextModel
| |-- layers[0..47](N=48): Gemma3DecoderLayer
| | |-- self_attn: Gemma3Attention
| | | |-- [q_proj, ..., o_proj](N=4): Linear
| | | |-- [q_norm, k_norm](N=2): Gemma3RMSNorm
| | |-- mlp: Gemma3MLP
| | | |-- [gate_proj, ...](N=3): Linear
| | | |-- act_fn: GELUTanh
| | |-- [input_layernorm,...](N=4):
    Gemma3RMSNorm
| | |-- norms: RMSNorm
| |-- rotary_emb: RotaryEmbedding
|-- lm_head: Linear
```

*Figure 5.* An example model tree for Gemma 3 model in vLLM.

**Model Analysis.** Raw model architecture is often excessively verbose, containing thousands of repetitive layers that can easily overwhelm an agent's context window. To resolve

this, EKKA parses and transforms the full model architecture into a *Model Tree*, a compressed form of the model architecture showing the hierarchical topology of the model. Figure 5 shows an example model tree representing model architecture of Gemma 3. Each node in the tree contains an identifier for the component (e.g. `self_attn`) as well as the class name of the component (e.g. `Gemma3Attention`). It abstracts away concrete layer indices and groups repeating sub-modules. This provides the agent with a concise, high-level map of the model architecture.

**Codebase Analysis.** An exact match on the component identifier or component class name does not necessarily indicate that they are equivalent. Figure 6 shows an example of the implementation of `RotaryEmbedding` in HuggingFace and vLLM respectively. The HuggingFace implementation is used for calculating the rotary embeddings themselves while the vLLM implementation is for applying rotary embedding to the hidden states. To resolve this, EKKA performs Codebase Analysis that statically parses the codebase of each framework into a code index, recording the file and line number defining each class as well as the class inheritance relationship. During component mapping, EKKA provides a tool `get_class_definition` to retrieve class definition from the code index, allowing it to check if the two components are truly equivalent.

**Incremental Mapping and Mapping Validation.** Generating a full component mapping in one shot is error-prone

**HuggingFace RotaryEmbedding forward definition**

```
1  def forward(self, x, position_ids):
2      # Calculates frequency based on position_ids
3      inv_freq_expanded = self.inv_freq.expand(...)
4      freqs = (inv_freq_expanded @
         position_ids_expanded).transpose(1, 2)
5      # Returns the embeddings (cos/sin) themselves
6      emb = torch.cat((freqs, freqs), dim=-1)
7      return emb.cos(), emb.sin()
```

**vLLM RotaryEmbedding forward definition**

```
1  def forward(self, positions, query, key=None):
2      # Retrieves pre-computed cos/sin from cache
3      cos_sin = self.cos_sin_cache.index_select(...)
4      cos, sin = cos_sin.chunk(2, dim=-1)
5      # Applies rotation directly to input tensors
6      query = apply_rotary_emb(query, cos, sin)
7      key = apply_rotary_emb(key, cos, sin)
8      # Returns rotated hidden states
9      return query, key
```

*Figure 6.* Comparison of the `forward` implementation of `RotaryEmbedding` in HuggingFace and vLLM.

```
1  def postprocess_hf_activations():
2      """Align HF separate QKV to vLLM fused QKV.
3      HF Shapes: q_proj (1,S,1024), k/v_proj (1,S,512)
4      vLLM Shape: qkv_proj (S,2048) """
5      # === [Fixed Template] Load Raw Traces ===
6      hf_q_raw = torch.load("../hf_q_proj_output.pt")
7      ...
8      vllm_raw = torch.load(...)
9      # === [Fixed Template] Create Result ===
10     result = {"self_attn.qkv_proj":
11         {"HF": None, "vLLM": None}}
12     # === [Agent Generated] Alignment Logic ===
13     # 1. Extract tensor and squeeze batch dim
14     # Shape becomes: (Seq_Len, Hidden_Dim)
15     hf_q = hf_q_raw[0].squeeze(0)
16     hf_k = hf_k_raw[0].squeeze(0)
17     hf_v = hf_v_raw[0].squeeze(0)
18     # 2. Concatenate along last dim
19     # (Seq_Len, 1024 + 512 + 512) -> (S, 2048)
20     hf_aligned = torch.cat([hf_q, ...], dim=-1)
21     # Prepare Final Result
22     result[...]["HF"] = (hf_aligned,)
23     result[...]["vLLM"] = (vllm_raw[0],)
24     return result
```

*Figure 7.* Example alignment code generated from the code template to align QKV projections between HuggingFace and vLLM.

due to the complexity of model architecture. Instead, EKKA generates component mapping iteratively and incrementally. At each iteration, a Mapping Validator checks if the components in the mapping are valid components in the Model Tree to improve mapping accuracy. If the validation passes, mapped components are removed from the Model Tree in the next iteration, otherwise EKKA will provide error feedback to fix the invalid mapping. To ensure a full component mapping, the Mapping Validator additionally checks if all components in the two frameworks are either mapped or provided a reasoning for why they are not mapped.

### 3.3. Activation Alignment

Even for semantically equivalent components, direct activation comparison is difficult due to implementation differences. For example, aligning HuggingFace Q/K/V projections with vLLM requires concatenating Q/K/V outputs to match vLLM's fused `QKVProjection`. Errors introduced in such postprocessing step is likely to cause inaccuracy in comparing the outputs of a component pair.

EKKA frames activation alignment as code generation: it produces executable Python postprocessing logic. It first collects each component's output activations during Trace Collection, and provides the agent a sketch of each output (dtype/shape and a few sample values) to guide alignment code generation. EKKA further improves alignment accuracy using helper tools for tensor matching and a knowledge base of validated alignment examples.

Figure 7 shows an example for aligning QKV projections, where the agent removes HuggingFace's extra batch dimension and concatenates outputs along the hidden dimension.

The code template handles loading raw traces and packaging the output format, so the agent only needs to implement the core postprocessing logic.

**Helper Tools.** Due to implementation difference, the activations that need to be aligned may appear at different indices in the outputs collected. EKKA provides useful helper tools that help infer tensor matches. `get_tensor_sum` takes two tensors A and B and returns their sum respectively, since sum match is a strong indication of tensor match. `infer_tensor_match` samples a few random elements from tensor A, find indices in tensor B whose values differ by a threshold, and return matches. `get_class_definition` from Component Mapping stage is also available to check class implementations.

**Knowledge Base.** To further improve accuracy of the generated alignment code, EKKA collects example alignment code of each component pair from correct implementation of the models. The models are first evaluated on downstream accuracy benchmarks and are verified that their performance has no gap between the target framework and the reference framework. Similar to the bug diagnosis pipeline, EKKA then provides example input, collects activations, and runs through Component Mapping and Activation Alignment to obtain alignment code examples. The alignment code in the knowledge base will be used as in-context examples during actual bug diagnosis, retrievable via tool-calling.

**Code Validation.** During bug diagnosis, a Code Validator validates the alignment code generated to ensure that the output format is correct (i.e., both target and reference framework produces a tuple of tensors with the same shape). Any format error detected will produce an error message

as a feedback to regenerate the alignment code. During construction of the knowledge base, the Code Validator additionally requires that error ratio (see Section 3.4) of the output activation pairs to be lower than a tunable threshold.

### 3.4. Error Analysis

Distinguishing implementation bugs from numerical precision effects is non-trivial. Models often run in low precision (e.g., BF16/FP8), where floating-point instability can accumulate across layers. Consequently, naively calculating the L2-norm or absolute error could be suboptimal, as error bound is highly component-dependent and unknown apriori.

Inspired by prior work (Jiang et al., 2026), EKKA introduces a robust error ratio that normalizes by the reference framework's precision error. Let $X_T$, $X_R$ be the outputs from the target and reference frameworks in low precision and let $Y_R$ be the output from the reference framework in full precision (e.g., FP32); the error ratio $\mathcal{R}$ is defined as:

$$\mathcal{R} = \frac{\|\mathbf{X_T} - \mathbf{Y_R}\|_2}{\|\mathbf{X_R} - \mathbf{Y_R}\|_2 + \epsilon} \qquad (1)$$

The denominator captures precision-only error in the reference, while the numerator captures additional deviation from the target (implementation + precision).

During Trace Collection, EKKA also records the call sequence of executed sub-modules. It then runs the alignment code obtained from the previous stage for each component along the sequence to compute error ratios, and applies change-point analysis to identify the earliest component where the error ratio becomes persistently elevated. Finally, EKKA produces a diagnosis report indicating the potential buggy components as well as the reasoning for why the component is suspected to be buggy.

**Change-point Analysis.** Identifying the buggy component is complicated by error propagation and alignment noise. To resolve this, EKKA applies a sustained elevation heuristic inspired by classic change-point detection methods (Page, 1954; Adams & MacKay, 2007): it searches for the earliest component that triggers a permanent upward shift in the error ratio, rather than a transient spike. Unlike alignment noise which typically manifest as a single outlier that immediately returns to baseline levels, a true buggy component causes the error ratio to remain consistently elevated across a window of subsequent operations. By prioritizing the first point of sustained divergence, EKKA effectively isolates the root cause component from downstream propagation.

### 4. Implementation

EKKA implements its core agent-based diagnosis using LangGraph (LangChain AI, 2026). Trace Collection em-

ploys an Activation Collector based on PyTorch forward hooks to capture activations and call sequences for all sub-modules, writing results to disk after generation. The collector is lightweight and portable, requiring under 100 lines of code to adapt to most PyTorch-based inference frameworks.

For error analysis, we apply an empirical error-ratio threshold of 1.5, observing that semantically correct implementations across frameworks consistently remain below this bound, while larger ratios indicate deviations beyond numerical noise. This threshold effectively filters transient spikes while remaining sensitive to sustained divergences caused by real bugs. EKKA currently localizes root causes at the PyTorch module level. We believe the principles of our differential debugging approach could be extended to achieve function- or line-level granularity. However, such an advancement would require implementing a more granular trace collection tool capable of automatically capturing every intermediate variable within the `forward` function, rather than relying solely on module-level output hooks.

In terms of extensibility, although EKKA uses PyTorch hooks for Trace Collection and PyTorch module hierarchies for Model Analysis, the workflow itself is not tied to PyTorch. For example, in JAX/Flax stacks (Bradbury et al., 2018), activations can be collected with `nnx.capture`, while module hierarchies can be collected using utilities such as `nnx.display(model)` or `nnx.iter_modules(model)`. The core diagnosis pipeline of component mapping, activation alignment, and error analysis remains unchanged.

### 5. Evaluation

The evaluation of EKKA mainly answers four questions: 1) What is EKKA's accuracy at diagnosing existing silent errors in LLM serving frameworks? 2) How do components in EKKA's design contribute to silent error diagnosis? 3) What is the cost of running EKKA? 4) Can EKKA diagnose new silent errors in LLM serving frameworks?

### 5.1. Silent Error Benchmark

We curated a benchmark of 17 real-world silent errors collected from two widely used serving frameworks, vLLM and SGLang. We first assembled a 9-bug benchmark from the bug study for initial evaluation, and then extended it to 17 silent errors. As detailed in Table 1, these cases represent a diverse set of open-sourced models and root causes in model and kernel implementation, ranging from logic errors in kernels, distributed inference bugs to numerical precision mismatches. We built docker containers for the benchmark containing reproduced silent errors from the two frameworks as well as runnable bug reproduction scripts.

We prioritize silent errors with confirmed root causes in the

| Issue | Symptom | Root Cause |
|---|---|---|
| vLLM-15393 | e5-mistral-7b-instruct result is inconsistent with HF | cumsum in BF16 accumulates errors |
| vLLM-16296 | Llama 4 accuracy regression under tensor parallel | RMSNorm dimension mismatch |
| vLLM-17689 | Gemma 3 degraded accuracy compared to HF | Sliding window applied to all layers |
| vLLM-23804 | Qwen 3 Reranker accuracy loss under tensor parallel | Final score layer is incorrectly sharded |
| vLLM-25833 | ERNIE 4.5 accuracy regression | Gate and bias use incorrect dtype |
| vLLM-26812 | MambaMixer 2 produces inconsistent results | Mamba output returned incorrectly |
| vLLM-33560 | Qwen3-NVFP4 accuracy regression | Data overflow for Marlin NVFP4 kernel |
| SGLang-4434 | Llama 1B W8A8_FP8 accuracy regression | FP16 weights loaded without quantization |
| SGLang-4807 | Gemma 3 generates garbage outputs | Sliding window size is incorrectly set |
| SGLang-7936 | Llama 4 generates garbage outputs | Paged attention is not fully supported |
| SGLang-10138 | Qwen 3 MoE w8a8 produces incorrect outputs | Incorrect intermediate size in fused expert |
| SGLang-10344 | Qwen 2 accuracy regression | Dual stream implementation error |
| SGLang-13044 | Qwen 2 model output is inconsistent with HF | FlashInfer kernel |
| SGLang-17887 | Qwen3-VL model generates incorrect result | Incorrect weight loading of lm_head |
| SGLang-18358 | DeepSeek OCR 2 results differ from HF | Attention causal mask not updated |
| SGLang-21039 | Qwen3.5-4B produces garbled output with TP=2 | Argument mismatch in MLP forward calls |
| SGLang-21093 | Qwen3.5-4B result is incorrect when PP=2 | lm_head weights not loaded on PP rank 1 |

*Table 1.* Accuracy bugs used for evaluation of EKKA in vLLM and SGLang.

model stack that can be reproduced with reasonable effort. Among the 36 model stack issues in the 70-bug study, the benchmark includes 17 bugs: 12 drawn from that study and 5 reported after the study period. We excluded the remaining candidates for concrete reproducibility reasons, including hardware requirements (5 cases Blackwell-only, 2 cases ROCm-only), models too large for our hardware (9 cases), framework features outside EKKA's current setting (4 cases), and 4 cases requiring older framework versions or highly specific concurrency/sequence-length conditions.

Section 5.3, Section 5.4, and Section 5.5 use the 9-bug subset, while other experiments use the full benchmark.

## 5.2. End-to-end Accuracy

**Baselines.** We evaluate EKKA against state-of-the-art software engineering agents capable of repository-level code analysis and debugging: OpenCode (OpenCode, 2026): the state-of-the-art open source coding agent and Mini-SWE-Agent (Yang et al., 2024b): the state-of-the-art software engineering agent designed to navigate repositories and fix bugs via command-line interfaces. For each bug case, both baselines were provided with the original bug report, the codebase and model path, and a bug reproduction script. They were tasked to identify the root cause component but operated without EKKA's specialized diagnosis stages.

**LLM Backend.** We use Claude Sonnet 4.5 (Anthropic, 2026a) as the LLM backend for EKKA and all baselines for results reported in main evaluation section. Additional ablation study on LLM backend can be found in Section A.2.

**Metric.** For end-to-end diagnosis accuracy, we evaluate the pass@$k$ metric (Chen, 2021). For each bug case in the benchmark, we run EKKA and baseline systems for $k$ trials and calculate the number of times the systems detect the root cause component correctly. All systems we evaluated

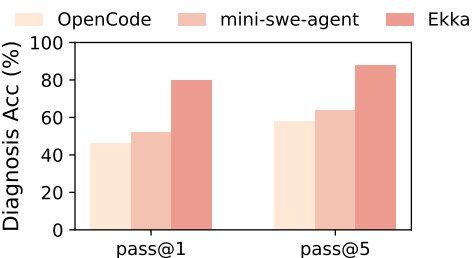

*Figure 8.* End-to-end diagnosis accuracy comparison.

are tasked to output the top-3 buggy component.

Figure 8 shows the end-to-end diagnosis accuracy comparison of EKKA against baselines over the pass@$k$ metric. EKKA achieves the best accuracy on both pass@1 and pass@5 metric. For pass@1, it shows 28% improvement over Mini-SWE-Agent and 34% improvement over Open-Code. For pass@5, it shows 24% improvement over Mini-SWE-Agent and 30% improvement over OpenCode.

## 5.3. Ablation Study - Component Mapping

Next we evaluate the effectiveness of techniques in component mapping stage. We compare three cases, 1) One-shot: generating complete component mapping in a single trial, 2) With Validation: generating component mapping incrementally with validation and error feedback 3) With Validation and Tool: setting in 2) with tool from Codebase Analysis. We measure *Mapping Accuracy*: the coverage percentage of ground-truth mapping in the mapping generated by the agent. Each bug case is run for 5 trials and we report the average mapping accuracy in Figure 9. Compared to one-shot setting, adding mapping validation improves mapping accuracy by 21.7%. Adding tool that retrieves class definition additionally improves mapping accuracy by 7.6%.

We also measure how component mapping design choices

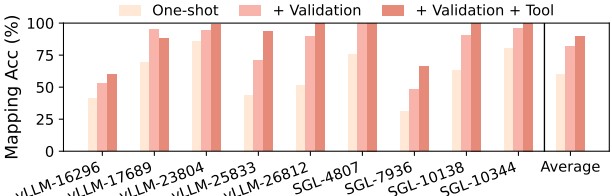

*Figure 9.* Ablation Study for Component Mapping.

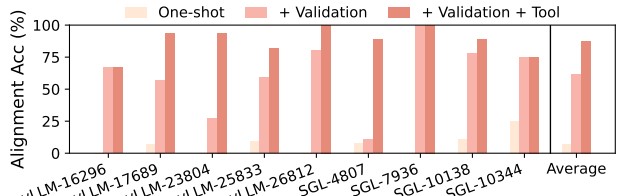

*Figure 10.* Ablation Study for Activation Alignment.

| Config | pass@1 | pass@5 |
|---|---|---|
| Full EKKA | 0.84 | 0.88 |
| CM w/o Tool | 0.67 | 0.77 |
| CM w/o Validation | 0.47 | 0.66 |
| AA w/o Tool | 0.39 | 0.55 |
| AA w/o Validation | 0.69 | 0.77 |

*Table 2.* End-to-end ablation study for component mapping (CM) and activation alignment (AA).

affect final diagnosis accuracy. The end-to-end results are summarized in Table 2. To isolate the role of component mapping, we keep activation alignment fixed using alignment code from a prior successful alignment. Removing tool for Codebase Analysis and incremental mapping reduces pass@1/5 from 0.84/0.88 to 0.67/0.77, while removing the mapping validation/refinement loop causes a larger drop to 0.47/0.66. This shows that both tool support and validation improve final accuracy, with validation contributing larger.

### 5.4. Ablation Study - Activation Alignment

For ablation study on activation alignment, we also compare three cases, 1) One-shot: generating alignment code for each component in a single trial, 2) With Validation: generating alignment code with code validation and error feedback 3) With Validation and Tool: setting in 2) with helper tools and knowledge base. We measure *Alignment Accuracy*: the percentage of components in the call sequence before the root cause component that are correctly aligned within the error threshold. This is because all components that are executed before the root cause component should be correctly implemented and thus have small error ratio. Each bug case is run for 5 trials and we report the average alignment accuracy in Figure 10. Compared to one-shot setting, adding code validation and error feedback improves alignment accuracy by 54.9%. Adding helper tools and knowledge base additionally improves alignment accuracy by 26%.

We also evaluate how activation alignment design choices affect final diagnosis accuracy in Table 2. To isolate the role of activation alignment, we use the ground-truth component mapping. Removing helper tools and the knowledge base lowers pass@1/5 to 0.39/0.55, while removing the alignment validation/refinement loop lowers pass@1/5 to 0.69/0.77. These results show that tool support and validation loops improve not only intermediate alignment quality but also final diagnosis accuracy, with tools and the knowledge base having the largest impact in this stage.

### 5.5. Sensitivity to the Error Threshold

To evaluate the sensitivity to the error threshold, we ran EKKA error analysis on the bug benchmark. For each bug case, we ran 5 independent trials and report pass@1 and pass@5 accuracy under different error thresholds.

The results in Table 3 show that diagnosis accuracy is stable across all three threshold values. Pass@5 is identical at all thresholds, and pass@1 varies by at most 0.04 (from 0.84 to 0.88). We also evaluated the error threshold distribution on non-buggy component pairs across both BF16 and FP8 settings. Their p50 ranges from 0.86 to 1.09 and p99 from 0.93 to 1.41, all below 1.5, which is consistent with the normalized definition of the error ratio and keeps correct components close to 1 across dtypes. Across ground-truth buggy components in the benchmark, the error ratio ranges from 3.13 to 1093.75. Full tables are provided in Section A.1. This supports the use of 1.5 as the default threshold: non-buggy component pairs remain well below this value across BF16 and FP8 settings, while ground-truth buggy components are well separated above it.

### 5.6. Cost Analysis

Table 4 summarizes the token usage statistics and estimated diagnosis cost across all benchmark cases. We measure input and output tokens for each LLM request and report averages over 5 runs. Using Claude Sonnet 4.5 as the backend, EKKA incurs an average cost of under $30 per diagnosis in the worst case without prompt caching. This shows that EKKA is a cost-efficient automated pipeline to integrate into the existing testing workflow of serving frameworks.

### 5.7. Newly Diagnosed Silent Errors

To demonstrate the practicality of the approach and show that EKKA does not overfit to the bug benchmark of reproduced silent errors, we use EKKA to diagnose open silent errors in vLLM and SGLang. Over the past month, EKKA diagnosed 4 new silent errors (2 in vLLM and 2 in SGLang). Table 5 shows the symptoms and root causes of newly diagnosed silent errors. We present the diagnosis report to the developers and all of them have been confirmed. Below shows a case study of a new bug diagnosed by EKKA.

| Threshold | pass@1 | pass@5 |
|---|---|---|
| 1.5 | 0.84 | 0.88 |
| 2.0 | 0.84 | 0.88 |
| 2.5 | 0.88 | 0.88 |

*Table 3.* End-to-end diagnosis accuracy under different error thresholds on the bug benchmark.

| Metric | Input Tokens | Output Tokens |
|---|---|---|
| Min | 1.63M | 150K |
| Max | 6.45M | 517K |
| Average | 4.03M | 304K |
| **Dollar Cost (avg)** | **$12** | **$17.6** |

*Table 4.* Token usage statistics across all benchmark cases.

**SGLang-16132.** A user reported that running Qwen Image on SGLang generates images with reduced quality. Compared to diffusers (von Platen et al., 2022) as the standard implementation, the text in the images generated from SGLang is garbled and contains logic errors. We run EKKA on the diffusion pipelines of the two frameworks with the prompts provided by the user. EKKA is able to align activations of components in the diffusion model and identifies that activations diverge after the normalization module in each denoising step. Figure 11 shows the code fix that replace the guidance rescaling normalization with standard L2-norm.

## 6. Related Work

**Software Fault Localization.** Fault localization techniques include *Spectrum-based* methods that correlate coverage with pass/fail outcomes (Wong et al., 2013; Li et al., 2018), *IR-based* methods that match bug reports to code (Zhou et al., 2012; Rahman & Roy, 2018), and *LLM-based* methods that predict faulty locations from code snippets (Kang et al., 2024; Chang et al., 2025; Yang et al., 2024a). These approaches often assume explicit failure signals, of which silent errors lack and thus are harder to localize. EKKA diagnoses silent errors effectively via differential debugging.

**Deep Learning Testing and Debugging.** Prior work generates diverse inputs to stress-test DL libraries (Wang et al., 2020; 2022) or performs differential testing across frameworks (Deng et al., 2023; Wang et al., 2025). These methods are often black-box or limited to high-level APIs, making it difficult to isolate internal state mismatches in optimized serving engines. Recent work targets silent errors but mainly for detection instead of diagnosis (Jiang et al., 2025; Suo et al., 2025; Jiang et al., 2026). EKKA localizes root causes via fine-grained cross-framework alignment.

**LLM Agents for Software Engineering.** LLM agents can solve repository-scale tasks (Yang et al., 2024b; OpenCode, 2026; Anthropic, 2026b) and have been extended to automated root cause analysis (Xu et al., 2025; Yu et al., 2025b; Wang et al., 2024; Chen et al., 2024). However, these approaches primarily target functional bugs or service interrup-

| Issue | Symptom | Root Cause |
|---|---|---|
| vLLM-28539 | Gemma 3 shows accuracy regression on GSM8K | BOS token is not applied |
| vLLM-30777 | whisper-large-v3 produces incorrect output | CUDA graph capture |
| SGLang-13044 | Qwen 2 model output is inconsistent with HF | FlashInfer kernel |
| SGLang-16132 | Qwen Image generates images in reduced quality | Normalization after denoising |

*Table 5.* New silent errors diagnosed by EKKA.

```
1  # denoising step
2  pred = self.transformer(...)
3  neg_pred = self.transformer(...)
4  - neg_std = neg_pred.std(...)
5  - pred_std = pred.std(...)
6  - pred_rescaled = pred * (neg_std / pred_std)
7  + pred_norm = torch.norm(pred, dim=-1)
8  + neg_norm = torch.norm(neg_pred, dim=-1)
9  + pred = pred * (neg_norm / pred_norm)
```

*Figure 11.* Code patch that fixes SGLang-16132 by implementing normalization after each denoising step using L2-norm.

tions characterized by explicit failure signals, such as error logs or metric anomalies. They lack the domain-specific scaffolding required to investigate silent errors.

## 7. Discussion

EKKA is most effective for silent errors that induce reproducible activation divergence and have a reference implementation to compare with. It is less suitable for engine orchestration or concurrency bugs, hardware corruption, and performance bugs where no stable divergence trace exists. Extending beyond module-level localization, broadening support beyond the current PyTorch-based prototype, and reducing diagnosis cost through smaller backbones or stronger caching are promising directions for future work.

## 8. Conclusion

We present EKKA, an automated system for diagnosing silent errors in LLM serving frameworks. EKKA first collects static and dynamic context via code analysis and trace collection, then applies a multi-stage agent-based diagnosis workflow. The system aligns semantically equivalent components across frameworks, generates executable code to compare intermediate activations, and uses a precision-aware metric to localize divergences. Evaluated on a benchmark of real-world bugs, EKKA achieves up to 34% higher diagnosis accuracy than state-of-the-art baselines, while remaining practical and cost-effective at about $30 per case and diagnosing 4 new bugs in vLLM and SGLang.

## Impact Statement

This paper presents work whose goal is to advance the field of Machine Learning. There are many potential societal consequences of our work, none which we feel must be specifically highlighted here.

## Acknowledgments

This work is supported in part by the NSF CAREER Award 2333885; PRISM, one of the seven centers in JUMP 2.0, a Semiconductor Research Corporation (SRC) program sponsored by DARPA; as well as generous donations from NVIDIA, AMD, Intel, and Arm.

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

| Target | Reference | Model | Dtype | p50 | p99 |
|---|---|---|---|---|---|
| vLLM | HF | Llama-3.1-8B | BF16 | 0.90 | 0.93 |
| vLLM | HF | gemma-2-2b | BF16 | 1.09 | 1.33 |
| vLLM | HF | Qwen2.5-7B | BF16 | 0.94 | 1.41 |
| SGLang | HF | Llama-3.1-8B | BF16 | 0.90 | 0.93 |
| SGLang | HF | gemma-2-2b | BF16 | 1.06 | 1.40 |
| SGLang | HF | Qwen2.5-7B | BF16 | 0.86 | 1.00 |
| SGLang | vLLM | Llama-3.1-8B | FP8 | 0.99 | 1.03 |
| SGLang | vLLM | gemma-2-2b | FP8 | 1.03 | 1.23 |
| SGLang | vLLM | Qwen2.5-7B | FP8 | 0.96 | 1.31 |

*Table 6.* Error-ratio statistics on non-buggy component pairs across BF16 and FP8 settings.

| Bug Case | Buggy Component | Error Ratio |
|---|---|---|
| vLLM-16296 | q_norm | 17.49 |
| vLLM-17689 | attn op | 17.00 |
| vLLM-23804 | final score | 1093.75 |
| vLLM-25833 | gate | 9.47 |
| vLLM-26812 | mamba | 183.30 |
| SGLang-4807 | attn op | 3.13 |
| SGLang-7936 | attn op | 410.96 |
| SGLang-10138 | experts | 56.70 |
| SGLang-10344 | gate_up | 143.24 |

*Table 7.* Error ratios of the ground-truth buggy components in the reproduced 9 bugs in the benchmark.

# A. Additional Evaluation Results

### A.1. Threshold Calibration Statistics

We evaluated the error-ratio distribution on non-buggy component pairs across both BF16 and FP8 settings in Table 6. Their p50 ranges from 0.86 to 1.09 and p99 from 0.93 to 1.41, all below 1.5, which is consistent with the normalized definition of the error ratio and keeps correct components close to 1 across dtypes.

In contrast, as shown in Table 7, the ground-truth buggy components in 9 reproduced bugs in the benchmark have error ratios from 3.13 to 1093.75, all well above 1.5. This separation further supports our threshold choice.

### A.2. Backend Sensitivity: Claude Haiku 4.5

Our main evaluation uses Claude Sonnet 4.5 to control experimental variability and keep the study focused on EKKA's diagnosis workflow rather than differences in backbone capability. To assess robustness under a weaker model, we additionally evaluate EKKA and the baselines with Claude Haiku 4.5 on the bug benchmark in Table 8. For each bug, we run 5 trials and report pass@1 and pass@5, matching the paper setup.

The results show that EKKA remains substantially stronger under a weaker backbone. Under Haiku 4.5, EKKA still achieves 0.67 pass@1 and 0.77 pass@5, while Mini-SWE-Agent and OpenCode reach only 0.31/0.66 and 0.27/0.55, respectively. The gap is especially clear on case vLLM-17689, where EKKA achieves 5/5 successful diagnosis, compared with 1/5 for Mini-SWE-Agent and 0/5 for OpenCode. This bug stems from a subtle implementation error in the attention module that applies sliding-window attention to all layers incorrectly. In this setting, weaker baselines often drift toward irrelevant code paths or superficial mismatches, whereas EKKA's pipeline keeps the diagnosis grounded in numerical evidence through component mapping, activation alignment, and error analysis.

# B. Prompt Template

### B.1. Component Mapping

| Method | Sonnet 4.5 pass@1 | Sonnet 4.5 pass@5 | Haiku 4.5 pass@1 | Haiku 4.5 pass@5 |
|---|---|---|---|---|
| EKKA | 0.84 | 0.88 | 0.67 | 0.77 |
| Mini-SWE-Agent | 0.60 | 0.66 | 0.31 | 0.66 |
| OpenCode | 0.48 | 0.66 | 0.27 | 0.55 |

*Table 8.* Backend sensitivity on the bug benchmark: Claude Sonnet 4.5 versus Claude Haiku 4.5.

---

**Simplified Prompt Template for Component Mapping**

**Task Summary:**
You are given PyTorch model architectures from two inference frameworks. Your goal is to construct a component-level mapping between them for differential diagnosis.

**Inputs (per iteration):**

- Reference model tree with remaining unmapped components: `{ref_model_tree}`

- Target model tree with remaining unmapped components: `{target_model_tree}`

- Reference code index: `{ref_index_path}`; Target code index: `{target_index_path}`

- Problem description: `{state['problem_description']}`

**Step-by-step Instructions:**

1. Output explicit mapping pairs; component names may differ.

2. Allowed mappings: one-to-one and one-to-many (e.g., operator fusion). Many-to-many mappings are not allowed; decompose into valid mappings.

3. Use variables (e.g., `i`, `j`) to represent repeated layer indices; for each variable, list all valid values.

4. When ambiguous, use `get_class_definition(class_name, index)` to compare implementations.

5. Start from leaf modules, then map intermediate modules; always use full paths with the `root` prefix.

6. Partial mappings are allowed per trial; remaining unmapped components and current mapping will be provided in the next iteration.

7. Every remaining component on each side must appear either in `component_mapping` or in `reasoning` (separately for reference and target).

**Required Output Format:**

```
1 component_mapping: [([ref_paths...], [target_paths...]), ...]
2 variables: {"i": [...], ...}
3 reasoning: {"ref": [...], "target": [...]}  # unmapped components + reasons
```

*Figure 12.* Simplified prompt template used by EKKA for component mapping.

## B.2. Activation Alignment

---

**Simplified Prompt Template for Activation Alignment**

**Task Summary:**
You are given activation snapshots for a mapped component pair from the target and reference frameworks respectively. Your goal is to write postprocessing code that transforms outputs from the reference framework to match outputs from the target framework, while preserving correctness.

**Inputs (per trial):**
- Component mapping: {ref_components, target_components}
- Activation snapshots: {ref_preview, target_preview}
- Relevant forward definitions/context: {forward_context}
- Code template: {function_signature}
- Optional tolerance: {error_threshold} (for obtaining activation alignment code example)
- Optional tools: get_alignment_code_example, get_class_definition, infer_tensor_match, get_tensor_sum

**Step-by-step Instructions:**
1. Inspect the provided forward definitions to understand what each framework returns.
2. Identify which outputs from the reference framework correspond to the outputs in the target framework, and implement only the core postprocessing logic.
3. Return the final result dict with matching shapes.

**Constraints / Notes:**
1. Do not fabricate placeholder tensors.
2. Ensure outputs have matching shapes before returning.
3. Do not compute error metrics inside the function.
4. You may add temporary debug prints, but remove them when asked.
5. You may edit the docstring, but do not change the function signature.

**Required Output:** Return the completed Python function inside a "python" code block.

**Code Template (example):**
```python
def postprocess_hf_activations() -> Dict[str, Dict[str, List[torch.Tensor]]]:
    """Post-process HuggingFace hidden states to match vLLM hidden states.
    Output: result[vllm_component] = {"HF": tuple(...), "vLLM": tuple(...)}
    """
    # === DO NOT CHANGE THE CODE BELOW ===
    # load the activations
    hf_<component>_<kind> = torch.load(<path>, weights_only=False)
    vllm_<component>_<kind> = torch.load(<path>, weights_only=False)
    # create the result dict
    result = {"<vllm_component>": {"HF": None, "vLLM": None}, ...}
    # === DO NOT CHANGE THE CODE ABOVE ===

    # TODO: implement core postprocessing logic
    ...

    # === DO NOT CHANGE THE CODE BELOW ===
    return result
    # === DO NOT CHANGE THE CODE ABOVE ===
```

*Figure 13.* Simplified prompt template used by EKKA for activation alignment. The agent fills in only the core postprocessing logic and the template handles loading raw traces and packaging outputs.

## B.3. Error Analysis

---

**Simpliefied Prompt Template for Error Analysis**

**Task Summary:**
You are given an aligned call sequence from one forward pass. Each entry is `(component_path, error_ratio)` or `(component_path, false)` (no reliable aligned ratio). Your goal is to identify the most likely root-cause buggy component in model/kernel implementation.

**Inputs:**
- Problem description: `{state["problem_description"]}`
- Aligned call sequence: `{call_sequence_aligned}`
- Numerical-noise threshold: `{error_threshold}`
- Output budget: top-$k$ where `k = {state["detection_top_k"]}`

**Definitions / Caveats:**
- `false` means alignment missing/failed.
- Large ratios can be false positives due to misalignment (spikes).
- A ratio > `{error_threshold}` is more likely beyond simple round-off.

**Core Rule:**
A true buggy component usually appears as a change point: after it, error ratios of subsequent modules become consistently higher, not a one-off spike.

**How to Reason:**
1. Treat the call sequence as ordered in execution time.
2. Ignore `false` entries when computing statistics, but keep them for locating where a change might have happened.
3. Find candidate change points where the median/mean error ratio shifts upward and stays elevated for multiple subsequent components.
4. Penalize spiky components whose neighbors return immediately to baseline.
5. Prefer the earliest plausible component in a sustained-elevation region (later components may inherit error).
6. Provide evidence by citing a small window of entries before/after each candidate.

**Required Output:** Return top-$k$ suspected components ordered by confidence and a concise reasoning summary.

**Suggested Output Schema (example):**

```
1  suspected_components: ["root....", ...]  # length k
2  reasoning: "... brief evidence-based explanation ..."
```

*Figure 14.* Simplified prompt template used by EKKA for final error analysis based on change-point analysis.

