# OpenReview forum: "Ekka: Automated Diagnosis of Silent Errors in LLM Inference"
_ICML.cc/2026/Conference — ICML 2026 regular_

### Official Review · Reviewer_pJpu · 2026-03-09

**Soundness:** 3
**Presentation:** 3
**Significance:** 3
**Originality:** 3
**Overall Recommendation:** 5
**Confidence:** 4

**Summary:**

This paper presents EKKA, an automated diagnosis framework for silent errors in LLM inference systems. The key idea is to formulate silent error diagnosis as a differential debugging problem: by comparing the internal execution states of a target inference framework against a trusted reference implementation, EKKA performs component mapping, activation alignment, and error analysis to identify the most likely root-cause component. The problem addressed in the paper is practical and important, especially for high-performance inference frameworks such as vLLM and SGLang, where model quality may degrade without explicit system crashes or error messages. Experimental results show that EKKA achieves good performance on a benchmark of real silent-error cases and outperforms general-purpose code agent baselines.

**Compliance With Llm Reviewing Policy:**

Affirmed.

**Final Justification:**

The authors addressed my concerns.  So, I have changed my score to 5.

**Key Questions For Authors:**

1. Since the 9 benchmark cases are manually curated rather than randomly sampled, could the authors clarify the inclusion and exclusion criteria, and provide more evidence that this benchmark is representative of real-world silent error distributions?

2. The average diagnosis cost is around $30 per case. Have the authors explored ways to reduce this cost, such as stronger prompt caching, lighter-weight models, fewer validation rounds, or selectively applying expensive models only to certain stages?

3. The experiments appear to use only Claude Sonnet 4.5. How well does EKKA transfer to other LLMs, and are the results consistent across models with different capabilities and cost profiles?

**Limitations:**

yes

**Strengths And Weaknesses:**

Strengths:

**1. Well-motivated and well-structured methodology**

The proposed method is well designed and aligns naturally with the debugging needs of silent errors. Instead of treating the entire debugging process as a black-box agent task, the paper decomposes it into several explicit stages, including component mapping, activation alignment, and error analysis, and combines these with differential testing. This decomposition makes the overall approach more interpretable and easier to understand, while also reflecting how practitioners would realistically debug such issues in practice.

**2. Good empirical performance**

The experimental results are promising. EKKA shows improvements over general software engineering agent baselines on the benchmark.

Weaknesses:

**1. Limited benchmark size and potential selection bias**

The experimental evaluation is based on only 9 bug cases, which is a relatively small sample size. Moreover, these 9 bugs are not randomly sampled; instead, they are manually curated from real GitHub issues and then reproduced to form the benchmark. While such curation is understandable in systems research, it also raises the possibility of benchmark selection bias. As a result, it remains unclear whether the reported performance would generalize to a broader and more representative distribution of silent errors.

**2. High diagnosis cost**

The reported cost is about $30 per case, which is nontrivial. Since the approach relies on multiple rounds of agent reasoning, validation, and alignment, the token usage may become a practical concern for large-scale adoption. That said, compared with the human time and engineering effort required to manually diagnose complex silent errors, this cost is still not unreasonable. Thus, this is more a question of deployment tradeoff than a fatal limitation.

**3. Evaluation only considers Claude Sonnet 4.5**

The evaluation appears to consider only Claude Sonnet 4.5 as the underlying model. This makes it difficult to judge how robust the method is across different LLM backbones.

---

> ### Author Rebuttal · Authors · 2026-03-31
>
> ## Concern 1: Limited benchmark size and potential selection bias
>
> Please kindly refer to our answer to Concern 1 of reviewer wBxa that addresses the benchmark size being small.
>
> In terms of bug selection for the benchmark, we prioritize accuracy bugs that can be reproduced with reasonable efforts. We started from our bug study of 70 closed issues with confirmed root causes. Among these 70, 36 have root causes of model implementation or kernel backend. Our updated benchmark of 17 accuracy bugs includes 12 of these 36 (the other 5 bugs were reported after we conducted the case study). The remaining 24 were excluded for concrete reproducibility reasons:
>
> - 5 require NVIDIA Blackwell GPUs
> - 2 require AMD/ROCm
> - 9 use models too large for our hardware (e.g. DeepSeek-V3.2, DeepSeek-R1, Llama-3.1-405B)
> - 4 depend on framework-level features outside Ekka’s current setting (e.g. CUDA graph)
>
> After accounting for these constraints, only 4 candidates remained:
>
> - SGLang-1400 and vLLM-8408 both depend on older framework versions and require additional porting.
> - vLLM-14058 manifests only when concurrent long-prompt requests are served and a generation crosses the LongRoPE transition point (around 4K tokens). We did not prioritize this because reproduction depends on recreating a specific combination of serving concurrency and sequence length configuration.
> - SGLang-4123 is a multi-image Qwen2.5-VL case that we could likely include. We did not prioritize it because the benchmark already contains Qwen-VL bugs with similar vision-language behavior, and Ekka performs well on those cases.
>
>
> ## Concern 2: High diagnosis cost
>
> Our pipeline is intended for diagnosis after an accuracy bug has already been detected, rather than as a heavy-weight step to run on every commit in a CI/CD pipeline. In practice,CI would use a lightweight accuracy test suite to monitor for regressions (e.g. https://github.com/vllm-project/vllm/pull/34772, https://github.com/sgl-project/sglang/issues/20847), and the diagnosis pipeline would be invoked only when tests fail. Therefore the appropriate comparison is between our diagnosis cost and the cost of manual root-cause analysis once a regression is found. The average cost of $30 per diagnosed bug is modest and highly competitive compared to the several engineer-hours required for manual diagnosis. In addition, the diagnosis can run overnight or asynchronously, allowing engineers to focus their effort on bug repair. As inference prices continue to fall, we expect the cost to decrease further over time.
>
> ## Concern 3: Evaluation only considers Claude Sonnet 4.5
>
> We thank the reviewer for pointing this out. Our original evaluation used Claude Sonnet 4.5 primarily to control experimental variability and keep the study focused on Ekka’s diagnosis workflow rather than differences in base-model capability. To directly address this concern, we additionally evaluate Ekka and the baselines with Claude Haiku 4.5, a smaller and faster model in the same family, using the original bug benchmark. We ran 5 trials for each bug and reported pass@1 and pass@5, similar to the paper setup.
> | Method | Sonnet 4.5 Pass@1 | Sonnet 4.5 Pass@5 | Haiku 4.5 Pass@1 | Haiku 4.5 Pass@5 |
>   |--------|:---:|:---:|:---:|:---:|
>   | Ekka | **0.84** | **0.88** | **0.67** | **0.77** |
>   | mini-swe-agent | 0.60 | 0.66 | 0.31 | 0.66 |
>   | OpenCode | 0.48 | 0.66 | 0.27 | 0.55 |
>
> The results show that Ekka remains substantially stronger under a weaker backbone. Under Haiku 4.5, Ekka still achieves 0.67 Pass@1 and 0.77 Pass@5, while mini-swe-agent and OpenCode reach only 0.31/0.66 and 0.27/0.55, respectively. The gap is especially clear on vLLM #17689 (Gemma 3), where Ekka achieves 5/5 successful detections, compared with 1/5 for mini-swe-agent and 0/5 for OpenCode. This bug stems from a subtle implementation error in the attention module that applies sliding window attention to all layers incorrectly. In this setting, weaker baselines often drift toward irrelevant code paths or superficial mismatches, whereas Ekka’s  pipeline keeps the diagnosis tied to numerical evidence through component mapping, activation alignment, and error analysis.

---

> > ### Author Rebuttal · Reviewer_pJpu · 2026-04-01
> >
> > Thanks. The response is good. I have changed my score to 5.

---

### Official Review · Reviewer_wBxa · 2026-03-11

**Soundness:** 3
**Presentation:** 3
**Significance:** 4
**Originality:** 2
**Overall Recommendation:** 4
**Confidence:** 3

**Summary:**

This paper presents EKKA, a system for diagnosing silent errors in LLM inference frameworks through differential debugging against a reference implementation. The idea is to compare a buggy serving framework against a trusted reference implementation, align intermediate components and activations across the two systems, and then identify where divergence begins. Experiments on reproduced bugs from two popular serving frameworks show improved diagnosis accuracy over general-purpose software engineering agents, and the paper also reports several newly diagnosed bugs.

**Compliance With Llm Reviewing Policy:**

Affirmed.

**Key Questions For Authors:**

When does the approach break, and what kinds of bugs are inherently difficult for it?

**Limitations:**

I did not found discussions on limitations of the work.

**Strengths And Weaknesses:**

Strength:

- The paper is well written and easy to follow.

-  The problem is important and timely. Silent inference failures are very difficult to diagnose, yet they matter a lot for real-world deployment.

- The proposed pipeline is well structured, and the differential-debugging formulation is intuitive and convincing.

- The paper studies real bugs rather than only synthetic cases.

- The empirical study of bug symptoms, root causes, and developer workflows adds value beyond the proposed method itself

Weakness:

- The benchmark contains only nine reproduced cases for the main quantitative evaluation, which makes it hard to judge robustness across models, architectures, and bug types.

- The diagnosis granularity is still limited to module level, which may reduce usefulness when developers need more precise localization.

---

> ### Author Rebuttal · Authors · 2026-03-31
>
> ## Concern 1: Benchmark size is small
> We have expanded our benchmark from 9 to 17 accuracy bugs from vLLM and SGLang. The 8 newly added accuracy bug cases are:
>
> | Issue | Symptom | Root Cause |
> |-------|---------|------------|
> | SGLang-4434 | Llama 1B W8A8_FP8 accuracy regression on GSM8k | FP16 weights loaded into FP8 tensors without actual quantization |
> | SGLang-13044 | Qwen 2 model output is inconsistent with HF | FlashInfer kernel |
> | SGLang-17887 | Qwen3-VL model generates incorrect result | Incorrect weight loading of `lm_head` |
> | SGLang-18358 | DeepSeek OCR 2 results produced by SGLang differ from Transformers | Attention causal mask not updated |
> | SGLang-21039 | Qwen3.5-4B produces garbled output with TP=2 | Positional argument mismatch in MLP forward calls, causing the MLP all-reduce to be silently skipped |
> | SGLang-21093 | Qwen3.5-4B result is incorrect when PP=2 | `lm_head` weights not loaded on PP rank 1 |
> | vLLM-15393 | e5-mistral-7b-instruct batch result is inconsistent with HF | MeanPool `cumsum` in BF16 accumulates rounding errors |
> | vLLM-33560 | Qwen3-NVFP4 accuracy regression on lambada_openai | Data overflow for Marlin NVFP4 kernel |
>
>
> The expanded benchmark now covers significantly more diversity along multiple axes:
>
> - **Data types:** FP16, BF16, W8A8_FP8, and NVFP4 quantization
> - **Model modalities:** language models, vision-language models (Qwen3-VL), OCR models (DeepSeek OCR 2), and embedding models  (e5-mistral)
> - **Root cause categories:** weight loading errors, kernel bugs, numerical precision issues (rounding error, data overflow), attention mask errors, and parallelism bugs (TP, PP)
>
> We evaluated the newly added cases for pass@1 and pass@5, and compared them with OpenCode and mini-swe-agent, similar to the evaluation setup in the paper. Below shows the end-to-end diagnosis accuracy results on the expanded benchmark of 17 accuracy bug cases.
>
> | Method | Pass@1 | Pass@5 |
> |-------|:------:|:------:|
> | OpenCode | 0.46 | 0.58 |
> | mini-swe-agent | 0.52 | 0.64 |
> | Ekka | 0.80 | 0.88 |
>
>
> Ekka maintains a substantial advantage over baselines on the expanded benchmark with a 28% improvement over mini-swe-agent and 34% improvement over OpenCode on pass@1, demonstrating that its effectiveness generalizes across the broader range of bug types, model architectures, and runtime configurations. The consistency of improvement from 9 to 17 cases also supports the robustness of our approach.
>
> We would also emphasize that each case is a realistic runnable accuracy bug rather than a simplified example or a static issue description. In every case, we reproduce the bug in the actual serving framework, set up a working reference run, and collect activations from the buggy serving framework. We think this gives a more realistic picture of the diagnosis problem, while still covering diverse data types, model types, root causes, and parallelism settings. More broadly, the benchmark can also support future work on better debugging and repair agents, not just benchmark final accuracy.
>
>
> ## Concern 2: Diagnosis granularity is at module level
> Please kindly refer to our answer for Concern 4 of Reviewer XxSj.
>
> ## Concern 3: Failure cases for Ekka
> Ekka is effective at diagnosing accuracy bugs in model implementations and kernel backends, which our study identifies as a substantial portion (around 50%) of real-world accuracy bugs and which motivated our design.
>
> In contrast, Ekka is less effective for bugs that do not produce a stable activation-level divergence. In particular:
>
> -  **Framework orchestration or scheduling bugs.** These bugs may arise only under multi-request or long-running execution conditions, depending on timing or system load, and may not correspond to an incorrect forward-pass computation for a fixed request.
> - **Hardware silent data corruption (SDC).**  SDC can appear as activation-level numerical divergence, but the true cause lies in hardware rather than software, so attributing the error to a model component would be misleading. This is not specific to Ekka: any software-level diagnosis tool can produce false positives when the underlying fault is hardware corruption rather than a software bug. In such cases, proper diagnosis requires complementary hardware-level evidence such as ECC logs or DCGM diagnostics.
> - **Performance bugs.** When outputs remain correct but throughput or latency degrades, such as from suboptimal kernel selection, missed fusion opportunities, or memory fragmentation, Ekka’s differential debugging is not the right tool: these cases require profiling-based analysis instead.

---

> > ### Author Rebuttal · Reviewer_wBxa · 2026-04-02
> >
> > Your rebuttal answered my questions very well. Thank you!

---

### Official Review · Reviewer_mTHQ · 2026-03-12

**Soundness:** 2
**Presentation:** 3
**Significance:** 3
**Originality:** 3
**Overall Recommendation:** 4
**Confidence:** 3

**Summary:**

This paper studies the problem of automated diagnosis of silent errors in LLM inference frameworks and proposes an automated diagnosis system called EKKA. An important problem the paper attempts to address is: as modern LLM inference systems become increasingly complex and deeply optimized, how can one automatically localize the root-cause module when no explicit runtime error is raised, yet output quality degrades? This problem directly affects the reliability, reproducibility, and production quality assurance of large-model inference frameworks.
The core idea of the paper is to formulate silent error diagnosis as a cross-framework differential debugging problem: by using a semantically correct reference framework as the baseline, and through component mapping, activation alignment, and change-point analysis based on an error ratio, the system identifies the first module at which persistent deviation occurs. Overall, this idea is reasonable. The paper also forms a relatively complete loop across problem motivation, empirical study, system design, and experimental evaluation.

**Compliance With Llm Reviewing Policy:**

Affirmed.

**Key Questions For Authors:**

none

**Strengths And Weaknesses:**

Strengths
1.	The problem is important and practically meaningful. Compared with traditional crash bugs, silent errors are more subtle and often more costly. They are indeed a worthwhile research direction in LLM serving frameworks. Through a systematic analysis of issues from vLLM and SGLang, the authors show that this problem not only exists in practice, but also exhibits cross-layer and multi-root-cause complexity.
2.	The method is relatively well designed. EKKA does not merely let an agent read code and search for bugs in a generic way. Instead, it introduces several strongly constrained intermediate steps, including component mapping, activation alignment, error ratio analysis, and change-point detection. These steps help the agent focus more precisely on output discrepancies across models, and therefore localize potential root causes in a more targeted manner.
3.	The end-to-end experimental results are reasonably convincing. The authors compare EKKA with general-purpose SWE agents and also provide ablation studies on two key submodules, component mapping and activation alignment. These results suggest that the performance gains come from the proposed diagnostic scaffolding rather than merely from using a stronger underlying LLM.
4.	The proposed method appears to have strong practical value. The paper does not stop at benchmark reproduction experiments. I searched for the issue log related to SGLang-16132 mentioned in the paper. In that log, there appears to be a user who provided key guidance related to closing the issue. Although this does not prove a direct connection to the paper, it is at least a real-world case, and the issue details do show some correspondence with the outputs described in the paper.
Weaknesses
1.	The benchmark is still relatively small, which limits external validity. Although the authors studied 70 closed issues earlier in the paper, only 9 cases are actually used in the end-to-end benchmark. For a systems paper, 9 cases may be acceptable as a proof of concept, but they are still insufficient to strongly support claims of generalization to a broad range of LLM serving bugs. In particular, most of the evaluated cases come from vLLM and SGLang, both of which are high-performance serving frameworks in the PyTorch ecosystem. Therefore, more evidence is needed to show whether the conclusions can generalize to a broader inference/runtime stack.
2.	The threshold choice appears somewhat heuristic. The authors use an empirical threshold of 1.5 to distinguish normal error from abnormal deviation, but this seems more like an engineering heuristic than a principled choice. Is this threshold stable across different models and tasks? How sensitive is the change-point heuristic to noise or to long-chain error propagation?
3.	There is a possible mismatch between EKKA’s localization granularity and the benchmark’s ground-truth granularity. The paper currently limits root-cause localization to the PyTorch module level, and the authors explicitly acknowledge that function-level or line-level localization is not yet supported. However, the evaluation does not sufficiently clarify whether the true root causes of the 9 benchmark silent errors all naturally lie at the module level. Several error descriptions in Table 1 seem closer to internal implementation logic, dtype handling, incomplete kernel support, or branch-logic bugs, rather than a defect that cleanly corresponds to a single module boundary. For finer-grained real-world errors, such as function-level, statement-level, or kernel-internal bugs, the paper does not explain how EKKA could further refine localization, nor how useful this coarse-grained localization would be in actual debugging and repair practice.
4.	The current ablation study is not yet fully convincing. The paper explicitly states in the evaluation that it aims to analyze the contribution of EKKA’s components to silent error diagnosis. However, the current results mainly show that the proposed designs for Component Mapping and Activation Alignment improve their respective intermediate subtasks, rather than directly demonstrating how these design choices affect final end-to-end diagnosis accuracy. Since EKKA’s final output also depends on downstream error analysis and the entire diagnosis pipeline, improvements in intermediate metrics do not automatically imply proportional improvements in root-cause localization performance. A more convincing evaluation would report end-to-end ablations, for example, showing how final pass@1 or pass@5 changes after removing incremental mapping, mapping validation, the alignment knowledge base, or helper tools. Otherwise, the current ablations mainly show that the local module designs are effective, but do not yet sufficiently quantify their true contribution to EKKA’s final diagnostic capability.

---

> ### Author Rebuttal · Authors · 2026-03-31
>
> ## Concern 1: Benchmark size is small
>
> Please kindly refer to our answer to Concern 1 of reviewer wBxa, where we expanded the benchmark of accuracy bugs in vLLM and SGLang from 9 to 17 bugs.
>
> Additionally, Ekka is not tied to the PyTorch stack in terms of  system design. The PyTorch-specific parts in our current prototype are mainly implementation details for collecting activations and extracting the model tree. In our current prototype, these are implemented as PyTorch hooks and parsing printed module hierarchy. The core diagnosis pipeline stays the same: component mapping, activation alignment and error analysis. For example, to extend Ekka’s support to frameworks in JAX stack, `nnx.capture` can be used to collect module outputs in a way similar to PyTorch forward hooks, and `nnx.display(model)` or `nnx.iter_modules(model)` can provide the model structure used to generate the model tree.
>
> ## Concern 2: The threshold choice appears somewhat heuristic
> Please kindly refer to our answer for Concern 4 of Reviewer XxSj.
>
> Moreover, the change-point analysis is designed to handle both noise and long-chain propagation. It does not react to a single spike. Instead, it looks for the earliest point where the error remains consistently high along the call sequence. This filters isolated noise from imperfect mapping or alignment, while attributing the bug to the earliest sustained deviation rather than later downstream propagation.
>
>
> ## Concern 3: Localization granularity w.r.t root cause granularity
>
> We believe module-level localization is already highly useful in practice. Real serving framework bugs are often spread across model definitions, attention backends, weight-loading logic, and kernel wrappers across many files, so debugging without guidance requires substantial trial and error. In this setting, identifying the first module where behavior diverges can largely narrow the search space.
>
> To test whether this signal is actionable, we evaluate mini-swe-agent on bug fixing with and without Ekka’s diagnosis hint. In the baseline, the agent receives the original bug report, reproduction scripts, and source code. In the Ekka setting, it additionally receives a short hint describing the modules with elevated error and any clear structural pattern. We then measure whether the agent can produce a correct patch by rerunning the target accuracy benchmark and comparing against the reference framework. For example, for vLLM-23804 (incorrect sharding of the final score layer), Ekka provides the following hint:
>
> “All transformer layers (0–11), including their attention and MLP submodules, show consistent low error ratios (0.7x–0.85x). The massive spike occurs specifically at the score component at the end of the model (1093x).”
>
> Below shows the bug fix performance of the agent under two settings.
>
> | Setting | pass@1 | pass@5 |
> |---------|:------:|:------:|
> | Fix agent w/o Ekka hint | 0.27 | 0.33 |
> | Fix agent w/ Ekka hint | 0.82 | 0.88 |
>
> Overall, these results show that Ekka’s module-level diagnosis is already effective for downstream fixing. Providing the hint improves the fix agent from 12/45 to 37/45 pass@1. The main benefit is that the hint directs the agent to the right module, which is especially important when the root cause is difficult to find through direct code inspection or ad hoc testing.
>
>
> ## Concern 4: End-to-end performance for ablation study
>
> We added an end-to-end ablation study measuring how key techniques in component mapping (CM) and activation alignment (AA) in Ekka affect localization accuracy. We evaluate four ablated variants of Ekka:
>
> - CM w/o Tool: removes source-code inspection and incremental mapping.
> - CM w/o Validation: removes the mapping validation/refinement loop.
> - AA w/o Tool: removes tensor-inspection helper tools and the knowledge base.
> - AA w/o Validation: removes the alignment validation/refinement loop.
>
> To isolate the role of each stage, for CM ablations we keep activation alignment fixed using alignment code from a prior successful alignment, and for AA ablations we use the ground-truth component mapping. This allows us to attribute end-to-end performance directly to the design choices rather than error propagation from other stages. We run each variant for 5 trials on the original benchmark and report end-to-end diagnosis accuracy using pass@1 and pass@5.
>
> | Config | pass@1 | pass@5 |
> |--------------|:------:|:------:|
> | Full Ekka | 0.84 | 0.88 |
> | CM w/o Tool | 0.67 | 0.77 |
> | CM w/o Validation | 0.47 | 0.66 |
> | AA w/o Tool | 0.39 | 0.55 |
> | AA w/o Validation | 0.69 | 0.77 |
>
> We find that removing validation in component mapping causes the largest drop, reducing pass@1 from 0.84 to 0.47 and pass@5 from 0.88 to 0.66. For activation alignment, the biggest drop comes from removing tools and knowledge base, which lowers pass@1/pass@5 to 0.39/0.55. Overall, both tool support and validation loops materially improve final diagnosis accuracy, not just intermediate metrics.

---

### Official Review · Reviewer_XxSj · 2026-03-13

**Soundness:** 4
**Presentation:** 4
**Significance:** 4
**Originality:** 4
**Overall Recommendation:** 4
**Confidence:** 3

**Summary:**

The paper introduces EKKA, an automated diagnosis system designed to identify and localize "silent errors" in Large Language Model (LLM) serving frameworks. Silent errors are defects where the framework processes requests without crashing or producing explicit error signals, but the output quality silently degrades. The authors frame this as a differential debugging problem, leveraging a semantically correct reference implementation (like HuggingFace) to compare against a buggy target (like vLLM or SGLang) to detect bugs.

**Compliance With Llm Reviewing Policy:**

Affirmed.

**Final Justification:**

The rebuttal addressed my main concerns.

**Key Questions For Authors:**

- The empirical error-ratio threshold is set to 1.5. How sensitive is the final diagnosis accuracy to this specific hyperparameter? Does it need to be manually tuned for different data types (e.g., FP8 vs. BF16)?

See weaknesses.

**Limitations:**

The authors did not discuss the limitations of this work.

**Strengths And Weaknesses:**

Strengths:
- Strong motivation. The authors collect 90 real-world silent errors from prominent frameworks like vLLM and SGLang. Identifying that 43.8% of these issues manifest as end-to-end accuracy regressions, indicating it is indeed a bottleneck problem in LLM deployment.
- Solid ablation study. Figures 9 and 10 cleanly decompose contributions from validation feedback and tool augmentation for both mapping and alignment stages.
- Real-World impact. The authors found 4 previously unknown bugs in production-grade frameworks.

Weaknesses:
- Cost for continuous integration. The paper notes an average cost of roughly $30 per bug diagnosis using Claude Sonnet 4.5. While highly cost-effective for diagnosing a known regression, it may be too expensive to run continuously on every commit as part of a standard CI/CD pipeline.
- Reliance on perfect reference alignments. The method assumes the existence of a perfectly semantically equivalent reference implementation. If a target framework implements a highly experimental, custom optimization that has no conceptual equivalent in HuggingFace, EKKA's component mapping might fail or produce noisy mappings.
- Diagnose at Pytorch module level. EKKA currently localizes root causes only at the PyTorch module level (Section 4). While helpful, developers still need to manually inspect the module to find the specific line of faulty code.

---

> ### Author Rebuttal · Authors · 2026-03-31
>
> ##  Concern 1: Cost for continuous integration
> Please kindly refer to our answer to Concern 2 of reviewer pJpu.
>
> ## Concern 2: Reliance on perfect reference alignments
> We believe it is practical to assume the existence of a reference implementation. HuggingFace (HF) is the de facto platform for major open-weight LLMs, so model vendors have strong incentives to release models there for broad adoption. More importantly, Ekka does not depend on HF specifically. If an implementation is not available on HF, Ekka can still use another semantically equivalent reference, whose correctness can be established empirically, for example by confirming that it reproduces the expected benchmark accuracy of the model.
>
> Moreover, Ekka does not require every component to admit a perfect one-to-one match. During component mapping, components that exist only in one framework are allowed to remain unmapped with an explicit reason. For the mapped components, Ekka further reduces noise through activation alignment validation and the use of helper tools. Incorrect mapping or alignment mistakes may still create noise in the error ratio, but error analysis is robust by looking for the earliest point where the error ratio is persistently elevated, rather than reacting to single spikes.
>
> ## Concern 3: Diagnose at Pytorch module level
> Please kindly refer to our answer to Concern 3 of reviewer mTHQ.
>
> ## Concern 4: Sensitivity to error-ratio threshold
> To test sensitivity to the error-ratio threshold, we ran Ekka error analysis on the original 9-bug benchmark from vLLM and SGLang. For each (bug, threshold) pair, we ran 5 independent trials with Sonnet 4.5 and report pass@1 and pass@5 below.
>
> | Threshold | pass@1 | pass@5 |
> |-----------|:------:|:------:|
> | 1.5 | 0.84 | 0.88 |
> | 2.0 | 0.84 | 0.88 |
> | 2.5 | 0.88 | 0.88 |
>
>
> The results show that diagnosis accuracy is stable across all three threshold values. Pass@5 is identical at all thresholds, and pass@1 varies by at most 0.04 (from 0.84 to 0.88). This suggests that choosing 1.5 is a conservative setting rather than a fine-tuned one. The only consistently failing case is vLLM-16296, where the QKV projection module before the root cause QK norm is incorrectly aligned due to a permutation of model weights in vLLM, leading to a noise that affects detection result.
>
>
> We also evaluated the error-ratio distribution on non-buggy component pairs across both BF16 and FP8 settings. Their p50 ranges from 0.86 to 1.09 and p99 from 0.93 to 1.41, all below 1.5, which is consistent with the normalized definition of the error ratio and keeps correct components close to 1 across dtypes.
>
> | Target | Reference | Model | Dtype | p50 | p99 |
> |------|-------|-------|-------|----:|----:|
> | vLLM | HF | Llama-3.1-8B | BF16 | 0.90 | 0.93 |
> | vLLM | HF | gemma-2-2b | BF16 | 1.09 | 1.33 |
> | vLLM | HF | Qwen2.5-7B | BF16 | 0.94 | 1.41 |
> | SGL | HF | Llama-3.1-8B | BF16 | 0.90 | 0.93 |
> | SGL | HF | gemma-2-2b | BF16 | 1.06 | 1.40 |
> | SGL | HF | Qwen2.5-7B | BF16 | 0.86 | 1.00 |
> | SGL | vLLM | Llama-3.1-8B | FP8 | 0.99 | 1.03 |
> | SGL | vLLM | gemma-2-2b | FP8 | 1.03 | 1.23 |
> | SGL | vLLM | Qwen2.5-7B | FP8 | 0.96 | 1.31 |
>
> In contrast, the ground-truth buggy components in our 9 reproduced bugs have error ratios from 3.13 to 1093.75, all well above 1.5. This separation further supports our threshold choice.
>
> | Bug Case | Buggy Component | Error Ratio |
> |----------|-----------------|------------:|
> | vllm-16296 | q_norm | 17.49 |
> | vllm-17689 | attn op | 17.00 |
> | vllm-23804 | final score | 1093.75 |
> | vllm-25833 | gate | 9.47 |
> | vllm-26812 | mamba | 183.30 |
> | sgl-4807 | attn op | 3.13 |
> | sgl-7936 | attn op | 410.96 |
> | sgl-10138 | experts | 56.70 |
> | sgl-10344 | gate_up | 143.24 |

---

> > ### Author Rebuttal · Reviewer_XxSj · 2026-04-02
> >
> > I thank the authors for the response. I am keeping my (positive) score.

---

### Decision · Program_Chairs · 2026-04-30

**Decision:**

Accept (regular)

**Comment:**

This paper presents EKKA, an automated diagnosis system designed for detecting silent errors in LLM serving frameworks such as vLLM and SGLang. Such silent errors are harder to detect because the system does not crash and the output quality degrades without any explicit errors. EKKA treats this as a differential debugging problem, by comparing the potential buggy target against a trusted source (e.g., Huggingface). It follows a pipeline of component mapping; activation alignment; and change-point analysis using an error ratio to identify the first module where persistent numerical deviation occurs.

The motivation of this work is quite strong with potential real-world impact, since silent errors accounts for nearly half of the accuracy regressions according to the study presented in this work. This is also recognized by all the reviewers. Initial concerns are raised on the modest size of the dataset, as only 9 is being used in the end-to-end evaluations, but the authors have augmented the initial set with 8 more examples. Another weakness of this work is the dependency on semantically equivalent reference implementations, which may not always be available. Some of these concerns are also clarified to some extent during the rebuttal phase, so I hope the authors can incorporate the discussions and new results in the next version of the paper.